# Keratinocytes can modulate and directly initiate nociceptive responses

Kyle M Baumbauer[†‡], Jennifer J DeBerry[†§], Peter C Adelman[†], Richard H Miller, Junichi Hachisuka, Kuan Hsien Lee, Sarah E Ross, H Richard Koerber, Brian M Davis, Kathryn M Albers*

Department of Neurobiology, Pittsburgh Center for Pain Research, Center for Neuroscience, School of Medicine, University of Pittsburgh, Pittsburgh, United States

*For correspondence: kaa2@pitt.edu

[†]These authors contributed equally to this work

Present address: [‡]School of Nursing, University of Connecticut, Storrs, United States; [§]Department of Anesthesiology, University of Alabama at Birmingham, Birmingham, United States

Competing interests: The authors declare that no competing interests exist.

**Abstract** How thermal, mechanical and chemical stimuli applied to the skin are transduced into signals transmitted by peripheral neurons to the CNS is an area of intense study. Several studies indicate that transduction mechanisms are intrinsic to cutaneous neurons and that epidermal keratinocytes only modulate this transduction. Using mice expressing channelrhodopsin (ChR2) in keratinocytes we show that blue light activation of the epidermis alone can produce action potentials (APs) in multiple types of cutaneous sensory neurons including SA1, A-HTMR, CM, CH, CMC, CMH and CMHC fiber types. In loss of function studies, yellow light stimulation of keratinocytes that express halorhodopsin reduced AP generation in response to naturalistic stimuli. These findings support the idea that intrinsic sensory transduction mechanisms in epidermal keratinocytes can directly elicit AP firing in nociceptive as well as tactile sensory afferents and suggest a significantly expanded role for the epidermis in sensory processing.

## Introduction

Cutaneous primary sensory afferents are the first in a chain of neurons that convert environmental stimuli into recognizable sensations of touch, heat, cold and pain. Sensory neurons are diverse in nature and exhibit unique chemical, morphological and electrophysiological properties that allow specific responses to applied stimuli. In response to stimuli, the skin produces neuroactive substances that are postulated to directly and indirectly modulate the activity of sensory fibers (*Groneberg et al., 2005*). These substances include glutamate (*Nordlind et al., 1993*; *Fischer et al., 2009*), ATP (*Cook and McCleskey, 2002*; *Inoue et al., 2005*; *Dussor et al., 2009*; *Barr et al., 2013*), acetylcholine (ACh) (*Grando et al., 1993*; *Wessler et al., 1998*), epinephrine (*Khasar et al., 1999*; *Pullar et al., 2006*), CGRP (*Hou et al., 2011*), neurotrophic growth factors (*Truzzi et al., 2011*) and cytokines (*Shi et al., 2013*). The skin also expresses ligand-gated (glutamate, ATP, nicotinic, muscarinic, 5-hydroxytryptamine, glycine and gamma-aminobutyric) and voltage-gated (sodium, calcium, transient receptor potential [TRP], potassium and cyclic nucleotide) ion channels and growth factor and cytokine receptors (*Olah et al., 2012*). The expression of neuroactivators and voltage and ion-gated channels indicates that complex autocrine and paracrine signaling between epithelial and neural tissues underlie sensory signaling (*Conti-Fine et al., 2000*; *Peier et al., 2002*; *Zhao et al., 2008*; *Atoyan et al., 2009*; *Dussor et al., 2009*).

It has been proposed that non-neuronal cells of the skin, specifically keratinocytes, contribute to the initial transduction process through regulated release of neuroactive substances (*Zhao et al., 2008*; *Dussor et al., 2009*; *Mandadi et al., 2009*; *Hou et al., 2011*; *Barr et al., 2013*). Testing this in an intact system has been difficult because the complexity in skin-nerve interactions prohibits isolation of the skin and neuronal output (a behavioral reflex or the pattern of axonal firing) since any natural stimulation (e.g., mechanical or thermal) simultaneously affects both keratinocytes and sensory neurons. To address this problem, mice with targeted expression of light-activated channelrhodopsin

**eLife digest** When a person touches a hot saucepan, nerve cells in the skin send a message to the brain that causes the person to pull away quickly. Similar messages alert the brain when the skin comes in contact with an object that is cold or causes pain. These nerve cells also help to transmit information about other sensations like holding a ball.

Scientists believe that skin cells may release messages that influence how the nerves in the skin respond to sensations. But it is difficult to distinguish the respective roles of skin cells and nerve cells in experiments because these cells often appear to react at the same time. Researchers have discovered that a technique called optogenetics, which originally developed to study the brain, can help. Optogenetics uses genetic engineering to create skin cells that respond to light instead of touch.

Baumbauer, DeBerry, Adelman et al. genetically engineered mice to express a light-sensitive protein in their skin cells. When these skin cells were exposed to light, the mice pulled away just like they would if they were responding to painful contact. This behavior coincided with electrical signals in the nerve cells even though the nerve cells themselves were not light sensitive. In further experiments, mice were genetically engineered to express another protein in their skin cells that prevents the neurons from being able to generate electrical signals. When these skin cells were exposed to light, the surrounding nerve cells produced fewer electrical signals.

Together, the experiments show that skin cells are able to directly trigger electrical signals in nerve cells. Baumbauer, DeBerry, Adelman et al.'s findings may help researchers to understand why some patients with particular inflammatory conditions are in pain due to overactive nerve cells.

(ChR2) can be used to determine the contribution of each cell type to cutaneous associated behavior (withdrawal reflex) and generation of afferent APs. For example, Ji and colleagues (*Ji et al., 2012*) showed that blue light stimulation of the skin of transgenic rats that expressed ChR2 in primary afferents under the Thy-1.2 promoter exhibited nocifensive type responses. Similarly, Daou et al. (*Daou et al., 2013*) showed light-induced behavioral sensitivity in mice in which the Nav1.8 promoter drove expression of ChR2 in a subset of primary afferents. In another optogenetic model, Maksimovic and colleagues directed ChR2 expression to the non-neuronal Merkel cells of the epidermis. Using an ex vivo electrophysiologic preparation they showed that blue light stimulation of the isolated skin elicited AP trains in slowly adapting type 1 (SA1) afferents, thus confirming the essential transducer role of Merkel cells in transmission of mechanical stimuli by SA1 tactile afferents.

To further examine how the epidermis and cutaneous afferents communicate we analyzed mice in which ChR2 was targeted to either sensory neurons or keratinocytes to determine the contribution of each cell type to cutaneous associated behavior (withdrawal reflex) and generation of afferent APs. Similar to Daou et al. (*Daou et al., 2013*), we found that light stimulation of the skin and activation of ChR2 in sensory afferents elicits robust nocifensive behaviors in mice. Remarkably, for mice that only express ChR2 in skin keratinocytes, light stimulation was also sufficient to generate nocifensive behaviors and regulate firing properties and evoke APs in specific subsets of cutaneous afferents, several which are known to activate in response to painful stimuli. In addition, expression of the chloride pump NpHR3.0 in keratinocytes significantly reduced AP firing in cutaneous afferents. These data indicate that Merkel cells are not unique in their ability to directly generate action potentials in sensory neurons and that light-mediated activation of keratinocytes is sufficient to engage an endogenous mechanism that can directly regulate cutaneous afferent firing.

## Results

### Summary of afferent properties measured using ex vivo intracellular and fiber teasing recordings

In these electrophysiological experiments we have recorded from 200 characterized cutaneous afferents (86 C-fibers, 37 Aδ, 77 Aβ) from the three different mouse genotypes (49 Prph-ChR2, 80 KRT-ChR2, 71 KRT-NpHR). The response properties to natural stimuli (pressure, heat, cold) for

the different fiber types can be summarized as follows: Aβ-LTMRs had mechanical thresholds from 5 to 10 mN (mean 5.5 mN), while Aδ-LTMRs thresholds ranged from 1 to 5mN, with a mean of 2.3 mN. For A-HTMRs, Aβ-HTMRs had mechanical thresholds ranging from 10 to 25 mN, with a mean of 17.5 mN; Aδ-HTMRs thresholds were 5–100 mN, with a mean of 26.7 mN. Cutaneous C-fibers showed a range of response properties, with mechanical thresholds from 5 to 50 mN (mean 23 mN), heat thresholds of 37–50°C (mean 44°C), and cold thresholds of 1–18°C (mean 11°C). No significant differences in these values were observed between genotypes.

## Activation of ChR2 in primary afferents produces nocifensive behaviors and action potentials in multiple types of primary afferents

We first determined the extent to which ChR2 activation in sensory neurons mimicked natural stimulation. Mice harboring a cre-responsive ChR2-YFP fusion gene in the Rosa locus (Ai32 mice) were crossed with peripherin (Prph)-cre mice to target ChR2 to unmyelinated and myelinated primary sensory neurons. The YFP tag allowed visualization of ChR2-positive projections in the skin and cell bodies in the dorsal root ganglion (DRG) of Prph-ChR2 mice (*Figure 1A,B*). Myelinated and unmyelinated fibers expressed ChR2 as indicated by ChR2-YFP-positive fibers in the skin (*Figure 1A*) and physiological recordings (*Figure 1D,E*). Behaviorally, all Prph-ChR2 mice (5 out of 5 mice tested) demonstrated robust light-induced tail-flick or hindpaw withdrawal in <30 ms in response to a 473 nm laser light flash, consistent with previous findings (*Grando et al., 1993*; *Daou et al., 2013*). Wildtype littermate mice (n = 5) were unresponsive.

We then used an ex vivo skin/nerve/DRG/spinal cord preparation (*Figure 1C*) (*McIlwrath et al., 2007*; *Lawson et al., 2008*) to characterize cutaneous afferent response properties in Prph-ChR2 mice (*Figure 1D,E*). ChR2 neurons responded to blue light pulses ranging from 39.7 mW (5–10,000 ms) to 0.7 mW (1000 ms pulse). Recordings were made from 49 characterized sensory neurons from 7 mice with 26 responders that included 1 A-fiber and 25 C-fiber nociceptors (identified based on their response to noxious mechanical or thermal stimuli) (*Table 2*). Among laser-responsive C-fibers, 21 responded to mechanical stimuli and of these, 14 responded to heat and/or cold stimuli. Four were classified as responding only to heat stimulation and 7 responded only to mechanical stimuli.

Activation of Prph-ChR2 afferents revealed complex intrinsic firing properties. A Prph-ChR2 Aδ-HTMR (A-delta-high threshold mechanoreceptor) exhibited a tonic response to mechanical stimulation whereas blue light evoked a phasic response (*Figure 1D*). In a CMHC nociceptor (C-fiber responding to mechanical, noxious heat and cold stimuli), suprathreshold light stimulation produced tonic firing whereas suprathreshold mechanical stimulation evoked a more phasic response (*Figure 1E*). Latency to first response to mechanical and light stimulation was similar. Peak instantaneous frequencies (IF) were significantly higher for suprathreshold mechanical stimulation, averaging 33.9 Hz for mechanical vs 8.6 Hz for light stimulation (@ 39.7 mW) for all mechanically responsive C-fibers. Interestingly, the average peak IF seen with laser light was similar to that seen in polymodal nociceptors (the majority of cutaneous afferents) in response to noxious heat (*McIlwrath et al., 2007*; *Lawson et al., 2008*). This raised the possibility that afferent-expressed ChR2 activation can evoke a 'baseline' response of putative nociceptors that reflects the intrinsic properties of these cells and that more naturalistic responses require collaboration of surrounding cells, including keratinocytes.

## Activation of ChR2 in keratinocytes produces nocifensive behaviors and action potentials in multiple types of primary afferents

To determine if keratinocytes contribute to afferent activation, mice that express ChR2-YFP (ChR2) specifically in keratinocytes were generated by crossing Ai32 mice with *Krt14* keratin Cre mice (KRT14-Cre). KRT-ChR2 mice exhibited robust expression of ChR2 in epidermal keratinocytes and hair follicles of hairy skin and basal and suprabasal keratinocytes of glabrous skin (*Figure 2A*). ChR2 expression does not occur in other dermal structures (vasculature, muscle) or in the DRG (*Figure 1B*). KRT-ChR2 mice also exhibited behavioral responses to blue light stimulation (*Figure 2B*, *Table 1*), but at lower frequencies and with greater latencies relative to Prph-ChR2 mice. The average withdrawal latency for KRT-ChR2 mice was 15.75 s ± 2.26 (SEM) (see *Video 1*), compared to the millisecond withdrawal responses exhibited by Prph-ChR2 mice. Testing was done in a blinded manner and all KRT-ChR2 mice responded at least one time out of 10 trials with laser stimulation restricted to a 30 s maximum. Measures on human skin using a thermistor showed a slight laser-induced increase in surface

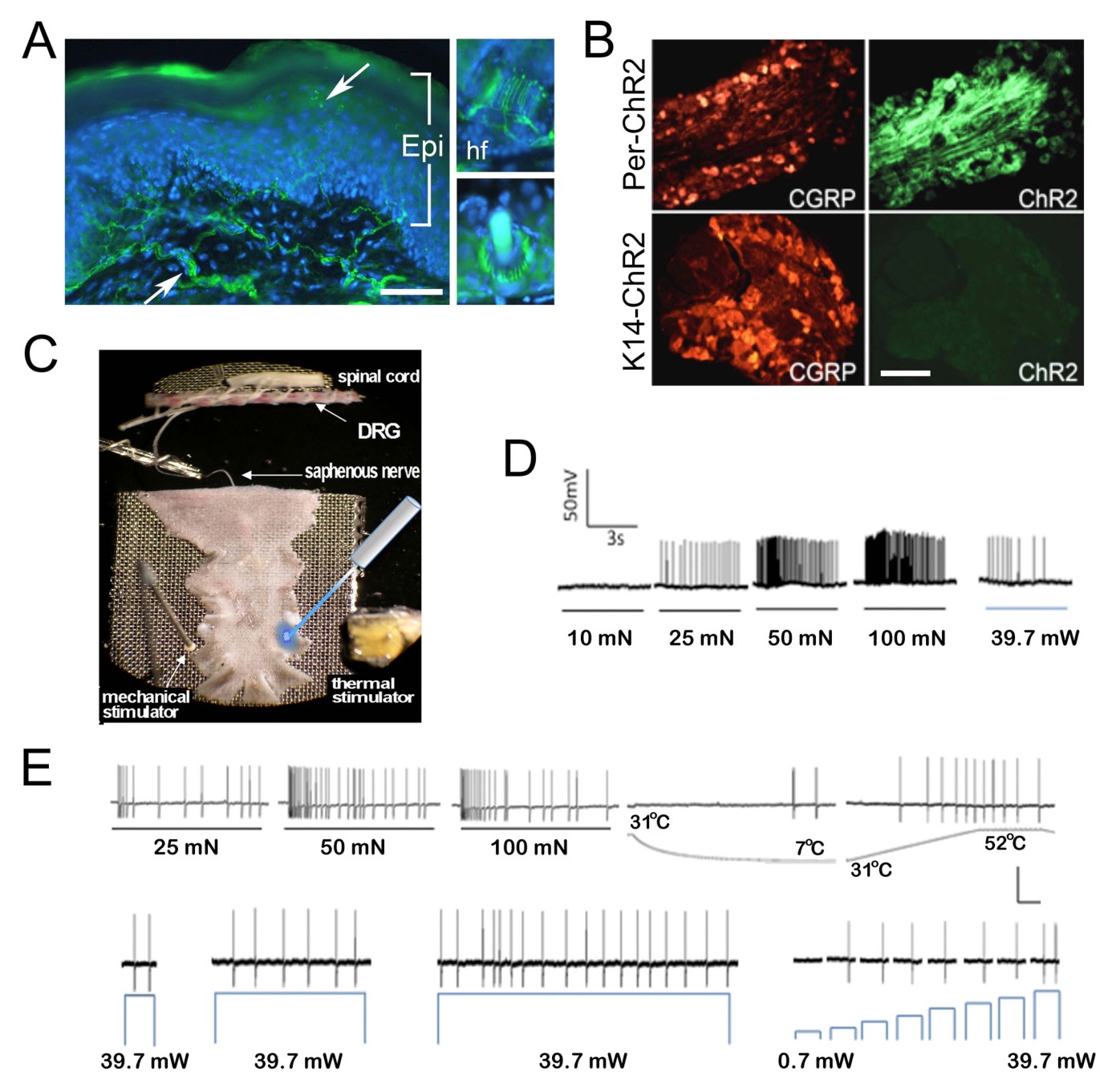

**Figure 1**. Light stimulates various types of cutaneous afferents in Prph-ChR2 transgenic mice. (**A**). ChR2-YFP expression in unmyelinated and myelinated (lanceolate endings of hair shaft, panels on right) fibers of Prph-ChR2 mouse skin. Arrows indicate nerve fibers in dermis and epidermis (Epi); DAPI (blue) labeling demarcates keratinocytes. (**B**). ChR2 is expressed in DRG neurons of Prph-ChR2 but not KRT-ChR2 mice. CGRP labels peptidergic neurons. (**C**). Ex vivo preparation used for functional characterization of cutaneous afferents in response to mechanical, heat and laser stimulation. (**D**). Response of a Prph-ChR2 Aδ-HTMR to mechanical and blue laser stimulation. (**E**). Recordings from a CMHC nociceptor from a Prph-ChR2 mouse in response to mechanical, thermal and light stimulation. Calibration bars in (**A**) = 250 μm, (**B**) = 100 μm, (**E**) = 60 mV/1 s, top trace; 40 mV/1 s, bottom trace.

temperature (from 27.5°C to 30.5°C) over the 30 s stimulation period, indicating that KRT-ChR2 mouse responses were not due to laser heating of the skin. That light activation of ChR2-keratinocytes could evoke nocifensive-type behaviors suggested that robust communication occurs between keratinocytes and sensory afferents that transmit nociceptive stimuli.

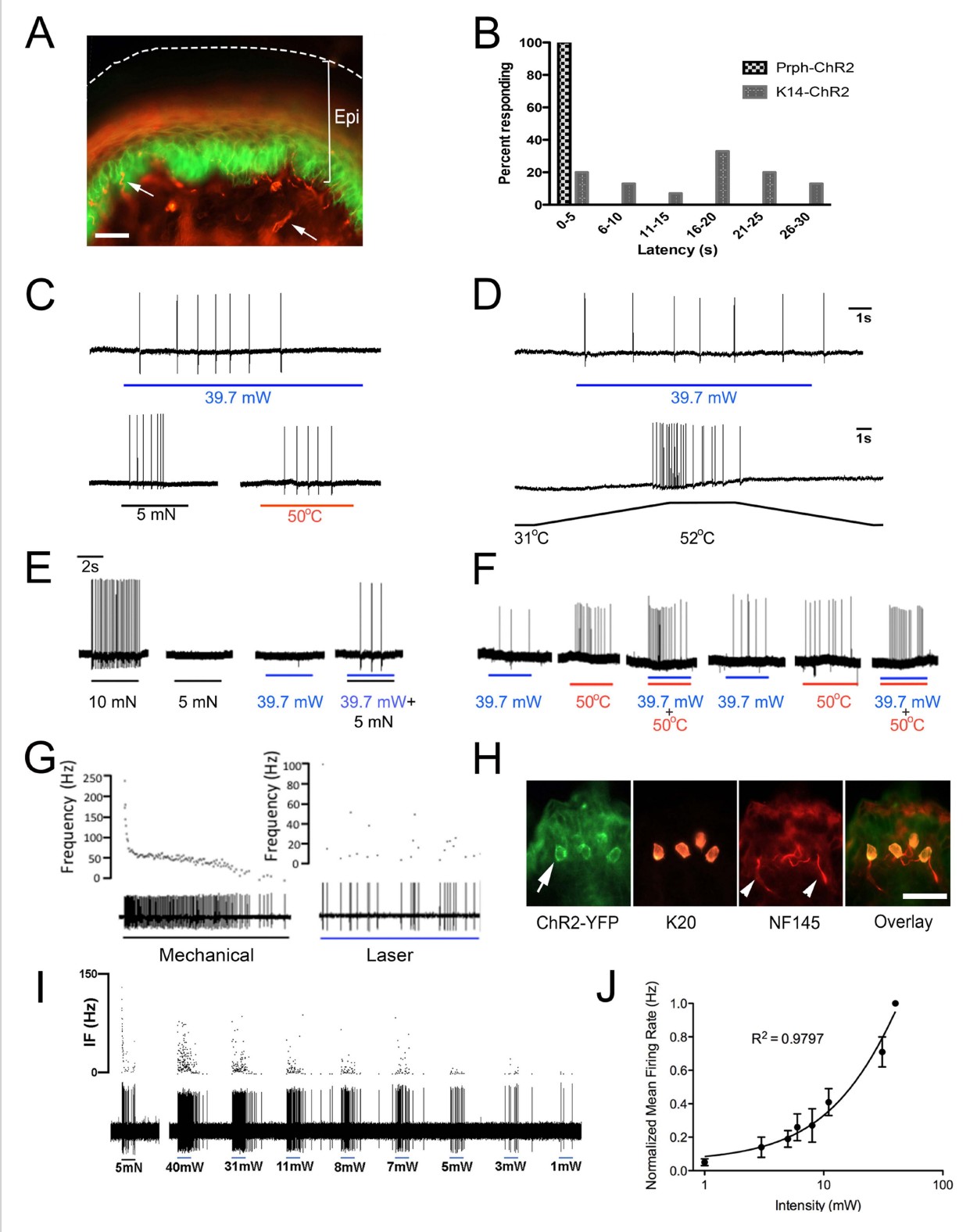

**Figure 2**. Blue light stimulates multiple subtypes of cutaneous afferents in KRT-ChR2 transgenic mice. (**A**). ChR2-YFP expression in keratinocytes of glabrous skin of KRT-ChR2 mouse. PGP9.5-positive nerve fibers (red) are in dermis and epidermis (arrows). (**B**). Plot of behavioral responses to blue laser across time intervals for Prph-ChR2 and KRT-ChR2 mice. All Prph-Cre mice showed an immediate response (within 5 s of stimulation). All KRT-ChR2 mice also responded at least once in 10 trials and with variable latencies (see *Table 1*). (**C**). Example showing activation of a CMH fiber type in response to blue

*Figure 2. continued on next page*

*Figure 2. Continued*

laser applied to KRT-ChR2 skin in the ex vivo preparation. Responses of this fiber to mechanical and heat stimuli are shown below laser response. (**D**). Example of a train of action potentials elicited in a CH fiber type in response to laser activation of the KRT-ChR2 skin. Responses of this fiber to heat stimuli are shown below laser response. (**E**). In this KRT-ChR2 Aβ HTMR afferent laser stimulation does not produce firing when presented alone, but does in combination with subthreshold (5 mN) mechanical stimulation. (**F**). Light directly activates this KRT-ChR2 CMHC fiber and summates with noxious heat stimulation. (**G**). SA1 Aβ-low threshold mechanoreceptor responds to mechanical and laser stimulation. (**H**). SA1s terminate on ChR2-YFP (green) positive Merkel cells co-labeled with anti-K20 (orange). Anti-NFH (red) labels SA1 fiber. Calibration bars in (**A**) and (**H**) = 100 μm. (**I**). Light-evoked responses from a SA-1 fiber at varying intensities (1–40 mW) with instantaneous frequency depicted. Pulses were 5 s in duration with 30 s between pulses. (**J**). Normalized mean firing rate vs light intensity plotted on a log-intensity scale. Data from 8 afferents are averaged from ascending and descending steps of light intensity, and were fit with a Boltzman sigmoidal function ($R^2$ = 0.98).

To further investigate keratinocyte-sensory neuron communication we used ex vivo preparations that employed both intracellular and fiber teasing recording techniques. Electrophysiological recordings were obtained from 80 cells isolated from 16 KRT-ChR2 mice (*Table 2*). Laser activation induced APs in 6 out of 24 unmyelinated nociceptive fiber neurons (*Figure 2C,D*) and in 4 out of 14 myelinated high-threshold mechanoreceptors (HTMRs) (not shown). These responses in heat-sensitive neurons are not due to laser-generated heat, as measures using a thermistor show minimal rise (~1 °C) in temperature over the 5 s recording interval. In addition, 3 myelinated HTMR fibers exhibited apparent summation when the laser was presented with natural stimuli. An example of this summation is shown in *Figure 2E*. This myelinated HTMR fiber had a mechanical threshold of 10 mN and neither a 5 mN mechanical stimulus nor the maximal intensity of blue light evoked a response. However, simultaneous application of 5 mN mechanical stimulation and light stimulation was sufficient to elicit APs. Recordings from 18 C-fiber nociceptors were maintained long enough to make multiple presentations of natural, laser and combined laser and natural stimuli. In 12 of these fibers, combined laser and natural stimulation evoked significantly more APs than natural stimuli alone (p < 0.01 paired T-test, n = 12) (*Figure 2F*). The remaining 6 C-fiber nociceptors did not display any summation when pairing laser and natural stimuli (not shown). Comparison of the functional properties of laser responsive and unresponsive nociceptive fibers revealed no significant differences. Laser activation also elicited AP firing in all 21 myelinated slowly adapting type 1 (SA1) low-threshold mechanoreceptors (LTMRs), which is most likely due to activation of Merkel cells (*Maricich et al., 2009*; *Maksimovic et al., 2014*), which, like epidermal keratinocytes, express the KRT14 keratin (*Figure 2G–J*). However, laser stimulation failed to activate any APs in myelinated rapidly adapting LTMRs.

AP firing following laser stimulation of keratinocytes was generally less robust than AP firing in Prph-ChR2 afferents (avg peak IF = 0.3Hz vs 8.6Hz, respectively). The exception was in recordings from SA1 fibers, which showed a robust, but atypical pattern of firing to light stimulation (*Figure 2G,I*). In response to mechanical stimulation SA1 fibers exhibit a characteristic response consisting of an initial high frequency burst of action potentials followed by a sustained firing, but at a lower frequency. Although light stimulation of these fibers could evoke high frequency bursts of activity, these bursts did not occur at

**Table 1**. KRT-ChR2 mice respond to blue light stimulation of paw skin

| Mouse strain | Sex | Responses/10 |
|---|---|---|
| KRT-ChR2 1 | Female | 4 |
| KRT-ChR2 2 | Female | 3 |
| KRT-ChR2 3 | Female | 1 |
| KRT-ChR2 4 | Male | 3 |
| KRT-ChR2 5 | Male | 1 |
| KRT-ChR2 6 | Male | 3 |
| Mean | | 2.5 |
| KRT-Cre | Male | 0 |
| KRT-Cre | Male | 0 |
| WT | Female | 0 |
| KRT-Cre | Female | 0 |
| WT | Female | 0 |
| Mean | | 0.0 |

All KRT-ChR2 mice respond to light applied to foot plantar skin whereas control littermates (n = 5) showed no response. The number of nocifensive responses (paw lifting, biting, licking) out of 10 stimulations was recorded. In total, light evoked responses in KRT-ChR2 mice in 17 of 60 total trials (28%). Control KRT-Cre mice lack the ChR2 gene whereas WT controls lack both transgenes.

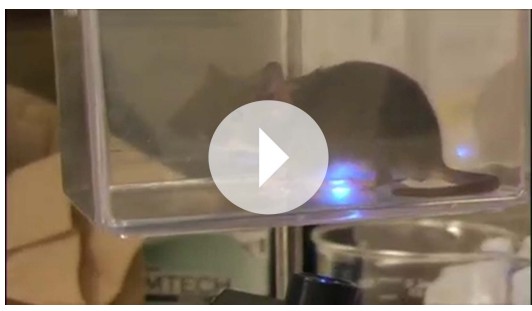

**Video 1.** KRT-ChR2 mice exhibit nocifensive behaviors in response to blue light. Blue light stimulation of channelrhodopsin expressing keratinocytes in the skin of KRT-ChR2 mice induces behavioral withdrawal responses. This mouse exhibits foot lifting at ~9 s after light exposure on the glabrous skin of the hind foot.

the initial onset of the light stimulus (mechanical mean peak IF = 218.2Hz; optical mean peak IF = 109.2 Hz) (*Figure 2G*). The SA1 response to light was stable, could be elicited repeatedly and was intensity dependent (*Figure 2I,J*).

## Keratinocytes from KRT-ChR2 mice are activated by blue light stimulation

To confirm that KRT-ChR2 keratinocytes are indeed activated by blue light, we examined the electrophysiological properties of these cells using whole cell patch clamp analysis. Keratinocytes do not normally generate APs, but they do have resting membrane potentials generated by currents mediated by ion (e.g., $K^+$, $Cl^-$) channels intrinsic to the plasma membrane (*Wohlrab et al., 2000*). Patch clamp recordings were made from keratinocytes isolated from adult tail skin of KRT-ChR2 mice

(*Figure 3A*). Recordings from 11 ChR2-YFP keratinocytes all showed inward current in response to a brief (1 s) flash of blue light (peak current: median 26.3 pA; steady current: 16.5 pA) (*Figure 3B,C*). No light-induced currents were recorded in keratinocytes cultured from wildtype mice (n = 4 cells).

## Activation of halorhodopsin in keratinocytes inhibits AP firing in cutaneous primary afferents

A loss of function approach using transgenic mice that express halorhodopsin (eNpHR3.0, 'NpHR') in keratinocytes was also used to further demonstrate the role of epidermal cells in afferent activation. Halorhodopsin is a yellow-to-red light-activated chloride pump that when expressed in neurons generates hyperpolarization, inhibits AP firing and neural activity (*Raimondo et al., 2012*). Using keratinocyte cultures from KRT-NpHR mice we recorded from 5 cells that all exhibited a hyperpolarizing response to orange light illumination. The median hyperpolarization was −1.1 mV. Using

**Table 2.** Number of primary afferents recorded from Prph-ChR2, KRT-ChR2 and KRT-NpHR mice that showed responses to light stimulation

| Cell type | Prph-ChR2 | | KRT-ChR2 | | KRT-NpHR | |
| --- | --- | --- | --- | --- | --- | --- |
| | Responsive | Unresponsive | Responsive (direct) | Unresponsive | Responsive | Unresponsive |
| SA1 | 0 | 3 | 21 (21) | 0 | 16 | 0 |
| RA (Aβ) LTMR | 0 | 4 | 0 | 15 | 0 | 9 |
| RA (Aδ) LTMR | 0 | 1 | 0 | 6 | 0 | 2 |
| A-HTMR (Aβ) | 1 | 1 | 3 (2) | 1 | 2 | 5 |
| A-HTMR (Aδ) | 0 | 2 | 4 (2) | 6 | 5 | 7 |
| CM | 7 | 0 | 1 (0) | 1 | 2 | 4 |
| CC | 0 | 2 | 0 | 1 | 0 | 1 |
| CH | 4 | 3 | 4 (2) | 3 | 0 | 1 |
| CMC | 0 | 1 | 1 (0) | 1 | 1 | 1 |
| CMH | 11 | 3 | 6 (2) | 0 | 7 | 5 |
| CMHC | 3 | 3 | 6 (2) | 0 | 2 | 1 |

Fibers that were activated directly by light stimulation of KRT-ChR2 keratinocytes are in parentheses.
Cell types recorded from are: SA1, slowly adapting type 1; RA (Aβ), rapidly adapting A beta low-threshold mechanoreceptor; RA (Aδ), rapidly adapting A delta low-threshold mechanoreceptor, A-HTMR, high-threshold mechanoreceptor(Aβ); A-HTMR, high-threshold mechanoreceptor (Aδ); CM, C mechanoreceptor; CC, C cold receptor; CH, C heat receptor; CMC, C mechano-cold receptor; CMH, C mechano-heat receptor; CMHC, C mechano-heat and cold receptor.

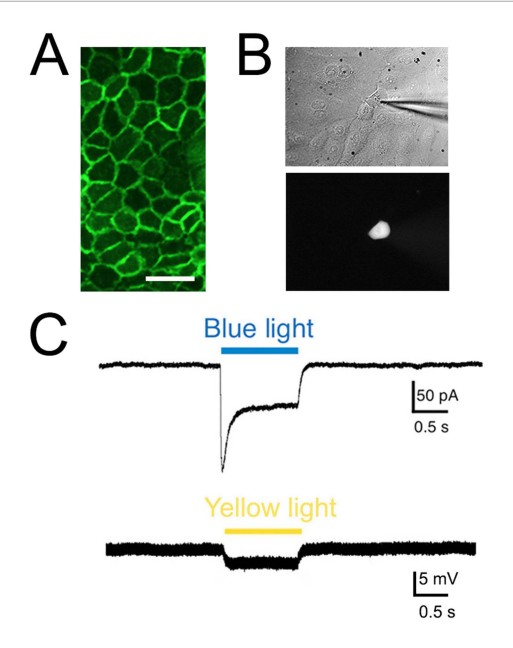

**Figure 3.** Light elicits current activation in cultured keratinocytes. (**A**). Fluorescent ChR2-YFP protein in plasma membrane of keratinocytes cultured from skin of KRT-ChR2 mice. (**B**). IR-DIC images of patch pipette on single keratinocyte that was recorded from and then filled with Alexa 555 dye. (**C**). Representative trace illustrates typical current evoked by blue light stimulation of KRT-ChR2. Yellow light stimulation of KRT-NpHR keratinocytes also produced a change in voltage properties of the cell. Control KRT-Cre keratinocytes that were isolated in parallel showed no response to light (not shown). Bar in A is 40 µM.

ex vivo preparations employing intracellular and fiber teasing techniques, 46 myelinated and 25 unmyelinated cells were recorded from 5 KRT-NpHR mice (*Table 2*). Application of yellow laser (589 nm) to the skin reduced AP firing in response to mechanical or heat stimulation in 12 of 25 C-fiber nociceptors and 7 of 19 myelinated nociceptors (*Figure 4*). This reduction was fiber type dependent with the most pronounced effects in mechanically sensitive C-fiber nociceptors (p = 0.02 Paired T-test n = 7) and slowly adapting type I LTMRs (p < 0.01 Paired T-Test n = 10) (*Table 2*). There were no effects observed on myelinated rapidly adapting LTMRs. It should also be noted that while in some presentations this yellow light-induced reduction in firing was 100% (*Figure 4A,B*), the average reduction in affected fibers was lower, that is, 44% in C fibers (n = 12), 48% in A-HTMRs (n = 7) and 44% in 16 SA1 fibers. In addition, in some cases where 100% reduction was observed, on subsequent light exposures the reduction in firing was less pronounced (*Figure 4B*).

## Discussion

These studies show in an intact skin preparation that ChR2-induced stimulation of skin keratinocytes, in isolation from other cells, is sufficient to induce AP firing in several types of sensory neurons. For some neuron subtypes, light activation of keratinocytes induces action potential firing similar to that evoked in response to natural stimuli. For other afferents, keratinocyte activation produced sub-threshold effects that potentiated the response to natural stimulation. For example, we recorded from afferents where light activation of keratinocytes alone did not elicit action potentials, but when combined with sub-threshold mechanical stimuli, produced multiple action potentials. These results suggest that keratinocytes are not only intimately involved in the generation of sensory neuron activity, but that the nature of this interaction is heterogeneous, differing for the many subtypes of sensory neurons that innervate the skin. Contributing to this heterogeneity may be the type or relative level of neuroactivator compound released by keratinocytes in response to mechanical, thermal or noxious stimulation or interactions with other cell types or structures in the skin, for example, immune cells or vascular structures.

Our electrophysiologic findings indicate that activation of Aδ and C fiber nociceptors likely underlies the behavioral sensitivity evoked by light in KRT-ChR2 mice. In addition, light stimulation of ChR2 expressed by Merkel cells likely transduces a signal that directly activates SA1 low threshold mechanoreceptors, as shown by Maksimovic (*Maksimovic et al., 2014*). That ChR2 in epidermal cells other than Merkel cells can activate numerous neuronal subtypes that are known to transmit thermal, mechanical and painful stimuli significantly expands the role of the epidermis in sensory processing.

The ability of keratinocytes to signal to sensory afferents and transmit pain is also supported by recent findings of Pang and colleagues (*Pang et al., 2015*). In these studies TRPV1 global knockout mice were genetically engineered to ectopically express TRPV1 selectively in keratinocytes. In these mice capsaicin could evoke nocifensive behaviors and c-fos expression in spinal cord dorsal horn neurons. As capsaicin application should only have activated keratinocyte-expressed TRPV1, it was concluded that these responses, which require activation of nociceptors, were initiated by keratinocytes, which in turn induced firing in primary afferents.

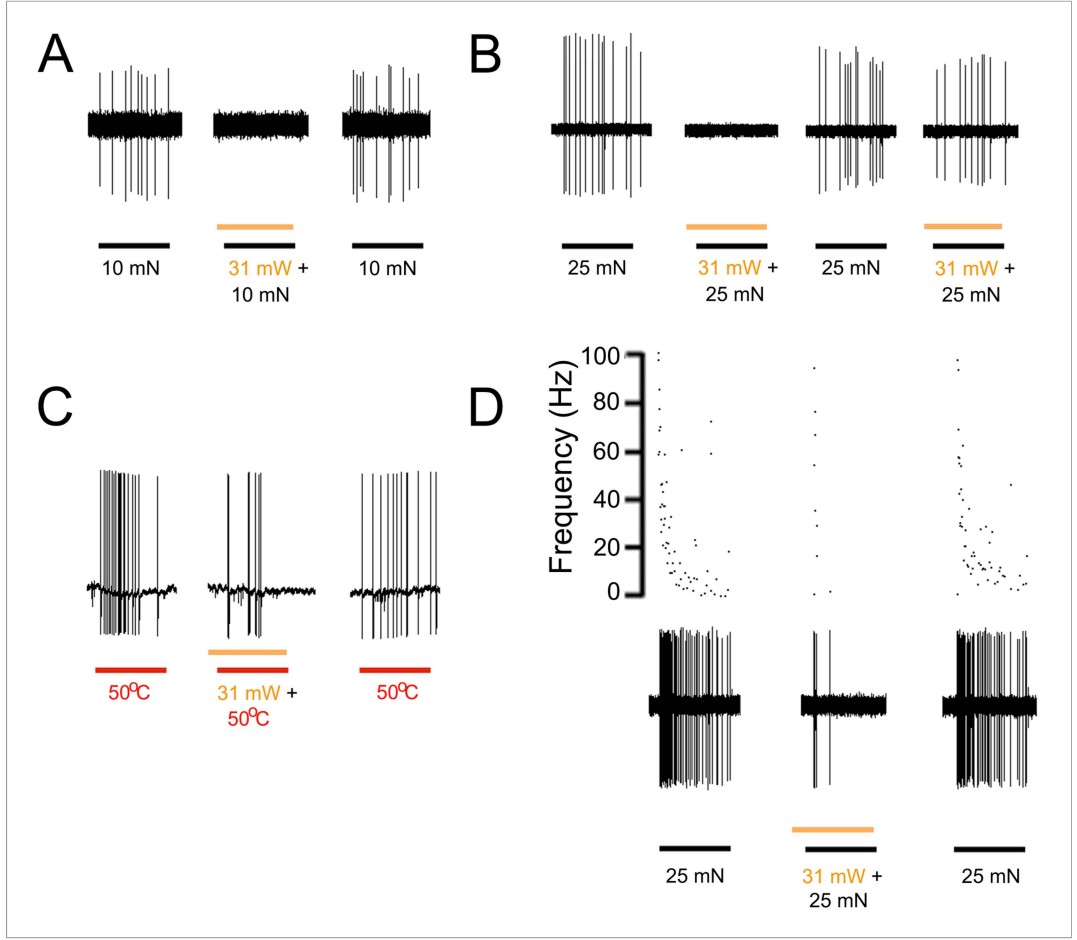

**Figure 4**. Yellow light inhibits AP firing in multiple subtypes of cutaneous afferents in KRT-NpHR mice. (**A**). Yellow light decreases AP firing in response to mechanical stimulation in this Aδ-HTMR afferent. (**B**). In this CMH-fiber the response to mechanical stimulation is decreased with the initial yellow laser stimulation; a smaller decrease in AP firing occurred with a second laser presentation. (**C**). This CMH-fiber showed decreased firing in response to heat in the presence of yellow laser stimulation. (**D**). Responses of a SA1 fiber to mechanical stimulation are significantly reduced by activation of NpHR in epidermal keratinocytes (which are likely Merkel cells). Laser stimuli (orange bars) occurred 1 s prior to mechanical (black bar) or heat (red bar) stimuli. Duration of each stimulus was either 5 s (mechanical and heat) or 6 s (laser).

Afferents that fire APs in response to light stimulation of keratinocytes were either polymodal, responding to mechanical and thermal stimuli, or unimodal, responding only to mechanical or thermal stimuli. For example, over half of the C-heat (CH) fibers, which only signal noxious heat and express TRPV1 (*Jankowski et al., 2012*), responded to keratinocyte activation. This suggests that keratinocytes have the ability to communicate directly with neurons that express TRPV1, an ion channel that transmits noxious heat and is required for inflammatory pain signaling (*Woodbury et al., 2004*; *Baumbauer et al., 2014*). Interestingly, LTMRs afferents, which form lanceolate endings around hair follicles (*Figure 1A*), were not activated by illumination of the skin in either Prph-ChR2 mice or KRT-ChR2 mice. A possible reason for this may be that these nerve fibers and/or the associated keratinocytes were not effectively illuminated due to the depth of the skin. However, in ongoing studies using Advillin-cre- and trkB-Cre[ER]-ChR2 mice, action potentials can be evoked in Aβ and Aδ LTMRs using the same light stimulus (not shown). Thus, it is possible that in Prph-ChR2 mice, an insufficient level of ChR2 for activation of LTMRs may exist. Another possibility is that the peripherin promoter only targets C-LTMR afferents. Unfortunately, the only cells we recorded from with lanceolate endings in these prreparations were myelinated RA-LTMRs.

In addition to the afferent stimulation, afferent activity could also be repressed by optogenetic stimulation of epidermal cells expressing NpHR. Light stimulation of NpHR and the predicted intracellular directed Cl flux led to significant reduction in many C-fiber, Aδ and SA1 afferent responses to mechanical and/or heat stimulation of the skin. Although the physiological and cellular mechanisms underlying this chloride-mediated change in keratinocyte signaling are yet to be resolved, the reduction in AP activity suggests a possible role for Cl⁻ in mediating neural-keratinocyte communication. Keratinocytes are known to exhibit chloride conductance (*Rugolo et al., 1992*), and Cl⁻ has been shown to contribute to changes in resting potential (*Wohlrab et al., 2000*) and keratinocyte hyperpolarization in response to mechanical stimuli evoked by hypotonic stress (*Gonczi et al., 2007*). Future studies, to determine if KRT-NpHR mice exhibit reduced behavioral responses in response to noxious stimuli, will require a system in which dual presentation of the stimulus, for example, heat and yellow light, are delivered.

The afferent responses evoked by keratinocyte stimulation were not at the same level evoked by natural stimuli, but this was not expected. It is most likely that keratinocyte activation is one contributor to natural stimuli-evoked sensory signaling (at least for some cells) and, in addition to neuronal activation, is a critical component of sensory transmission. Evidence for this is the clear activation of primary afferents by blue light stimulation of keratinocytes and the observed summation of AP firing in afferents exposed to light and mechanical or thermal stimuli. Importantly, physiological relevance is also indicated by the in vivo nocifensive behavior and clear withdrawal response elicited by light stimulation of KRT-ChR2 mice. These responses were much slower compared to behavioral response times measured in Prph-ChR2 mice, which express the ChR2 ion channel in the primary afferent. This difference may reflect the time needed for release by keratinocytes of neuroactivator compound(s) to a level sufficient to evoke an AP as well as the heterogeneity of fiber types innervating the epidermis. Further study of the types of neuroactivator compounds released by light stimulated ChR2 keratinocytes and the effect of these activators on specific types of primary afferents will address these issues.

Disturbances in epidermal-neuronal signaling in inflamed or damaged skin result in abnormal sensory transmission that underlies associated pain, itch and paresthesia (*Urashima and Mihara, 1998*; *Kinkelin et al., 2000*). The present findings support the idea that keratinocytes, as activators of cutaneous neurons, have a central role in the onset and maintenance of such abnormal transmission. These findings also suggest that altered release of keratinocyte expressed neuromodulators (e.g., ATP, CGRP), neurotransmitters (e.g., ACh) or activity of neurotransmitter receptors and ion channels could drive changes in transmission and importantly, may do so in a neuron subtype specific manner.

# Materials and methods

## Animals

Male and female mice ages 6–10 wks were used. Mice expressing ChR2 in sensory neurons were generated by crossing Ai32 mice with peripherin-Cre mice (*Zhou et al., 2002*), which were generously provided by Dr. Rebecca Seal (Department of Neurobiology, University of Pittsburgh). Transgenic mice that express ChR2 in keratinocytes were generated by crossing Ai32 mice (B6;129S-*Gt(ROSA) 26Sor^{tm32.1(CAG-COP4*H134R/EYFP)Hze}*/J ) with KRT14-Cre mice (Tg(KRT14-cre)1Amc/J), both obtained from Jackson Laboratories (Bar Harbor, ME). Mice expressing halorhodopsin (eNpHR3.0-EYFP) in keratinocytes were generated by crossing Ai39 mice (B6;129S-*Gt(ROSA)26Sor^{tm39(CAG-HOP/EYFP)Hze}*/J) with KRT14-Cre mice. All experiments were approved by the Institutional Animal Care and Use Committee at the University of Pittsburgh (protocol # 14074296).

## Immunocytochemistry

Skin and dorsal root ganglia were post-fixed in 4% paraformaldehyde, cryoprotected in 25% sucrose, embedded in gelatin, sectioned on a sliding microtome and labeled using target-specific antibodies followed by a fluorescently tagged secondary.

Sections were stained with antibodies to keratin K20 (1:20, mouse; Signet Covance, MA), NF145 (1:200, rabbit; Millipore, MA) or PGP9.5 (1:1000, rabbit; Ultraclone, UK) followed by appropriate secondary antibodies (Jackson ImmunoResearch) used at 1:500 dilution. Fluorescent images were captured using a digital camera attached to a Leica DM4000B fluorescence microscope (Leica, Wetzlar, Germany) and processed for brightness and contrast using Adobe Photoshop.

## Behavior

Laser-induced paw withdrawal latency was measured using an 80 mW, 473 nm wavelength laser from a distance of 8–10 mm while animals were confined in a glass container. For KRT-ChR2 and control mice the number of nocifensive responses (paw lifting, biting, licking) out of 10 stimulations was recorded.

## Ex vivo intracellular recording and fiber teasing

Comprehensive phenotyping of individual afferents was done using an ex vivo skin/nerve/DRG preparation as previously described (*McIlwrath et al., 2007*). Mice were anesthetized with ketamine/xylazine mixture (90/10 mg/kg, respectively) and perfused with oxygenated artificial cerebrospinal fluid (aCSF). The hairy skin of one hindpaw, saphenous nerve, DRGs, and spinal cord were dissected in continuity and placed in a bath of warm (31°C) circulating oxygenated aCSF. The skin was placed on an elevated metal platform exposing the epidermis to air for mechanical, thermal and laser stimulation. Electrophysiological recordings were performed by impaling individual neuronal somata using sharp quartz microelectrodes. Electrical stimuli were delivered through a suction electrode on the nerve to locate sensory neurons that innervate the skin. Receptive fields were localized and characterized based on responses to mechanical and/or thermal stimulation. Responsiveness to laser stimulation was determined using an 80 mW, 473 nm wavelength laser (to activate ChR2) or a 34 mW, 589 nm wavelength laser (to activate halorhodopsin)(Laserglow Technologies, Toronto, Canada) affixed to a micromanipulator. The distance from the skin was adjusted to produce a 1–2 mm diameter illuminated area. In the KRT-ChR2 experiments blue light and mechanical or thermal stimuli were applied simultaneously. The tip of the mechanical stimulator is 1 mm in diameter and typically did not block the entire receptive field available for laser stimulation. In addition, the light was delivered at a 45° angle, allowing penetration of the skin beneath the probe. In the KRT/HpHR experiments the yellow light preceded the natural stimulus by 1 s.

Neurons with conduction velocities $\leq$ 1.2 m/s were classified as C-fibers, while all others were classified as A-fibers. Fiber teasing experiments were performed using previously established protocols (*Zimmermann et al., 2009*) to further examine afferents in KRT-ChR2 and KRT-NpHR mice. The preparation was prepared in the same manner as the skin/nerve/DRG preparation, except the saphenous nerve was cut slightly proximal to the junction with the femoral nerve. Recordings were performed using a bipolar platinum electrode, and stimuli were administered to the epidermis.

## Culture of primary mouse keratinocytes

Adult mouse keratinocytes were cultured following the procedure of (*Redvers and Kaur, 2005*). Tail skin was digested in dispase II (8 mg/ml dissolved in HBBS containing 1% pen/strep) overnight at 4°C. The epidermal sheet was removed, digested in trypsin-ethylenediamine acid solution (Life Technologies, Waltham, MA) and the dissociated cells plated onto 12 mm glass coverslips coated with type 4 collagen at $10^4$ cells/coverslip. Cells were cultured in Keratinocyte Serum Free Medium (K-SFM, Life Technologies) supplemented with 0.1% pen/strep, 10 ng/ml epidermal growth factor and 0.1 nM cholera toxin. Patch clamp recordings were performed at 7–14 d post plating.

## Whole cell patch clamp electrophysiology

Whole cell patch clamp recordings were made on keratinocytes grown on coverslips exposed to a one second blue light pulse. Keratinocytes on coverslips were transferred to a recording chamber that was continuously perfused with extracellular bath solution containing (in mM): NaCl 140, KCl 5.4, $CaCl_2$ 1.8, $MgCl_2$ 1.0, HEPES (N-2-hydroxyethylpiperazine-N'-2-ethanesulfonic acid) 10.0 and D-glucose 11.1 (*Inoue et al., 2005*). The pH was adjusted to 7.4 with NaOH. Cells were visualized using a microscope with infrared differential interference contrast (IR-DIC) optics (Olympus, Pittsburgh, PA, BX-51WI). Patch pipettes made from borosilicate thin walled glass capillaries (Warner Instruments, G150F-6) using a P-97 micropipette puller (Sutter Instrument Company, Novato, CA) had a tip resistance of 10–15 MΩ. The composition of pipette solution was (in mM); 135 potassium gluconate, 5 KCl, 0.5 $CaCl_2$, 5 EGTA, 5 Hepes, 5 ATP-Mg, 0.025 Alexa 555, pH 7.2. All experiments were conducted at room temperature (19°C). Whole-cell patch clamp recordings were made using an Axopatch 200B amplifier (Molecular Devices, Sunnyvale, CA). The currents were clamped at −50 mV and a one second blue light pulse was delivered from a xenon light source (Lambda DG-4, Sutter Instrument Company) using a 40x water immersion objective and GFP filter set. Data were digitized using a Digidata 1322A (Molecular Devices) and stored and analyzed using pClamp 10 software (Molecular Devices).

## Acknowledgements

This work was supported by National Institutes of Health Grants T32 NS073548 to KMB and RHM, DK063922 to JJD, NIH AR063772 to SER, NIH NS075760 and NS050758 to BMD, NIH NS023725 to HRK and NIH NS033730 and AR066371 to KMA.

## Additional information

### Funding

| Funder | Grant reference | Author |
| --- | --- | --- |
| National Institute of Neurological Disorders and Stroke | T32 NS073548 | Kyle M Baumbauer, Richard H Miller |
| National Institute of Neurological Disorders and Stroke | NS075760 | Brian M Davis |
| National Institute of Diabetes and Digestive and Kidney Diseases | DK063922 | Jennifer J DeBerry |
| National Institute of Neurological Disorders and Stroke | NS050758 | Brian M Davis |
| National Institute of Neurological Disorders and Stroke | NS023725 | H Richard Koerber |
| National Institute of Neurological Disorders and Stroke | NS033730 | Kathryn M Albers |
| National Institute of Arthritis and Musculoskeletal and Skin Diseases | AR066371 | Kathryn M Albers |
| National Institute of Arthritis and Musculoskeletal and Skin Diseases | AR063772 | Sarah E Ross |

The funders had no role in study design, data collection and interpretation, or the decision to submit the work for publication.

### Author contributions

KMB, JJDB, PCA, Conception and design, Acquisition of data, Analysis and interpretation of data; RHM, JH, KHL, Acquisition of data, Analysis and interpretation of data; SER, Conception and design, Analysis and interpretation of data; HRK, BMD, KMA, Conception and design, Acquisition of data, Analysis and interpretation of data, Drafting or revising the article

### Author ORCIDs

Kyle M Baumbauer, http://orcid.org/0000-0003-0437-9209

### Ethics

Animal experimentation: This study was performed in strict accordance with the recommendations in the Guide for the Care and Use of Laboratory Animals of the National Institutes of Health. Animals were handled in compliance with an approved Institutional Animal Care and Use Committee (IACUC) protocol (#14074296) of the University of Pittsburgh. All surgery was performed under appropriate anesthesia with every effort was made to minimize pain.

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
