## [Decision Letter]

Thank you for submitting your work entitled “Keratinocytes can modulate and directly initiate nociceptive responses” for peer review at *eLife*. Your submission has been favorably evaluated by Gary Westbrook (Senior Editor), and three reviewers, one of whom, David Ginty, is a member of our Board of Reviewing Editors.

The reviewers have discussed the reviews with one another and the Reviewing Editor has drafted this decision to help you prepare a revised submission

This is an interesting manuscript that uses optogenetics to investigate the role of keratinocytes in modulating sensory neuron firing and subsequent behavioural responses. The paper is important from two angles, firstly at the technical level because it shows that light application to keratinocytes can indirectly activate sensory neurons, and secondly at a conceptual level showing that keratinocyte signalling may have a role in modulating sensory perception. The ChR2 experiments clearly show that optogenetic activation of keratinocytes can evoke activity in several classes of primary afferent neuron. While this had previously been shown for communication between Merkel cells and SA1 afferents, this paper demonstrates that it is also the case for many types of neuron such as A-HTMRs and heat responsive C-fibers, albeit at lower firing frequencies.

Essential revisions:

1) Figure 2 and Table 2 contain most of the relevant findings in the study and it would be useful if these were further expanded. For example in Figure 2, responses to natural stimuli (mechanical and/or heat stimuli) should be included for fibres in C and D. Similarly, in Table 2, the number of responsive neurons should be subdivided into those where a direct activation was observed or where the effect of blue light was modulatory. Are the A-HTMR fibres Aβ or Aδ units? This should be indicated. A summary table of afferent properties (conduction velocity, sensory thresholds, etc.) would strengthen the manuscript.

2) It is possible that afferents that form lanceolate endings (R-LTMRs and C-LTMRs) were not activated because they innervate hair follicles whose keratinocytes were not effectively illuminated due to their depth in skin. Does (*Prph*)-cre label these neurons (labeling of lanceolate endings alone does not address this, because C-LTMRs form lanceolate endings)? This is a potential limitation of the approach that warrants discussion.

3) Quantification needs improvement. For example, average values for the blue light and yellow light evoked responses in dissociated keratinocytes should be shown. Related to this, with the exception of Figure 2, the manuscript presents representative electrophysiological traces without providing summary data to illustrate the range of responses observed. Including a clear summary of the data will provide important information for other scientists who wish to build on the foundation presented here. This is particularly important for the data sets analyzed by paired *t*-tests (Figure 4).

4) Controls for genotype are provided for the behavior, but genetic line controls for the physiology experiments are not. We are concerned that three of the neurons that show robust responses are heat-sensitive; is heat from the laser activating these neurons? We would like to see a Cre- or Ai32- control showing that both *K14Cre* and *Ai32* alleles are necessary for the light-evoked response.

5) Behavioral experiments are also performed demonstrating that in some animals blue light activation of keratinocytes evokes nocifensive responses. These responses are often of long latency (mean 15 seconds) and only occur in 25% of mice. Figure 2 is therefore somewhat misleading as it shows latencies as a percent of responsive mice. It would be more accurate to present this as a percent of all mice for each genotype, thereby demonstrating that this is not a strong effect.

6) To explore whether keratinocytes have a role in the detection of natural stimuli, the authors have performed loss of function experiments using K14-NpHR. These are important experiments but are currently rather preliminary. For example, n numbers are low for some fibre types (CM and CC) or absent (CH, CMC, CMHC). Moreover it is not indicated what the threshold is for responsiveness in these experiments. The samples sizes and thresholds should be noted for these experiments, and additional experiments are needed in some cases to reach the appropriate numbers.

7) The authors have presented data using a single light pulse applied for durations of around 4 seconds. While analysis of the stimulus intensity versus firing rate is presented, it would also be important to show how stimulus duration and the frequency of light pulses can initiate activity. For example what is the minimum duration and number of pulses that can evoke neuronal activity when comparing direct neuronal stimulation versus keratinocyte stimulation? If such data are currently available, they would be a valuable addition to the study.

---

## [Author Response]

*1)*
Figure 2
*and*
Table 2
*contain most of the relevant findings in the study and it would be useful if these were further expanded. For example in*
Figure 2*, responses to natural stimuli (mechanical and/or heat stimuli) should be included for fibres in C and D. Similarly, in*
Table 2
*the number of responsive neurons should be subdivided into those where a direct activation was observed or where the effect of blue light was modulatory. Are the A-HTMR fibres Aβ or Aδ units? This should be indicated. A summary table of afferent properties (conduction velocity, sensory thresholds, etc.) would strengthen the manuscript*.

The requested data has been added to Figure 2 and incorporated into Table 2. A paragraph summarizing the afferent response properties to naturalistic stimuli (heat, cold and mechanical thresholds) has also been added to the beginning of the Results section (in the subsection “Summary of afferent properties measured using ex vivo intracellular and fiber teasing recordings”).

*2) It is possible that afferents that form lanceolate endings (R-LTMRs and C-LTMRs) were not activated because they innervate hair follicles whose keratinocytes were not effectively illuminated due to their depth in skin. Does (*Prph*)-cre label these neurons (labeling of lanceolate endings alone does not address this, because C-LTMRs form lanceolate endings)? This is a potential limitation of the approach that warrants discussion*.

In other ongoing studies using Advillin-cre/ChR2 and trkB-Cre^ER^-ChR2 mice we can evoke action potentials in LTMRs directly using the same light stimulus. Thus, we do not believe that the lack of activation of the lanceolate endings in the current study is due to a lack of illumination. Rather we suspect that the labeled lanceolate endings in the (*Prph*)-cre/ChR2 mice (Figure 1) are C-LTMR fibers. Unfortunately we did not record from any C-LTMR fibers in those experiments. It is also possible that in the (*Prph*)-cre/ChR2 mice the level of ChR2 expression is not sufficient to activate APs in these endings. These possibilities are now presented in the Discussion.

*3) Quantification needs improvement. For example, average values for the blue light and yellow light evoked responses in dissociated keratinocytes should be shown. Related to this, with the exception of*
Figure 2*, the manuscript presents representative electrophysiological traces without providing summary data to illustrate the range of responses observed. Including a clear summary of the data will provide important information for other scientists who wish to build on the foundation presented here. This is particularly important for the data sets analyzed by paired* t*-tests (*Figure 4*)*.

We have included the median current amplitude and hyperpolarization seen in these experiments in the Results section.

*4) Controls for genotype are provided for the behavior, but genetic line controls for the physiology experiments are not. We are concerned that three of the neurons that show robust responses are heat-sensitive; is heat from the laser activating these neurons? We would like to see a Cre- or Ai32- control showing that both* K14Cre *and* Ai32 *alleles are necessary for the light-evoked response.*

We have measured the amount of heat generated at the skin in the in vivo preparation during these stimulations and found very minimal heating associated with the laser illumination. The output of a thermistor in contact with skin exposed to laser shows that skin temperature increases 1°C from the baseline of 30°C. In the recording experiments the lowest heat threshold of the afferent recorded was 37°C. Thus, it is not possible that the fibers were responding to heat generated by the application of the laser light.

*5) Behavioral experiments are also performed demonstrating that in some animals blue light activation of keratinocytes evokes nocifensive responses. These responses are often of long latency (mean 15 seconds) and only occur in 25% of mice.*
Figure 2
*is therefore somewhat misleading as it shows latencies as a percent of responsive mice. It would be more accurate to present this as a percent of all mice for each genotype, thereby demonstrating that this is not a strong effect*.

We apologize for this confusion. All K14-ChR2 mice were responders. Each mouse responded at least one time out of 10 trials with laser stimulation restricted to a 30s maximum (as done in Hargreave’s testing). The plot in Figure 2 is to illustrate the distribution of the latencies, e.g., 20% of the responders did so between 0-5 seconds, 10% of responders did so between 6-10 seconds, etc. We have modified the text in the Results section to clarify these outcomes.

*6) To explore whether keratinocytes have a role in the detection of natural stimuli, the authors have performed loss of function experiments using K14-NpHR. These are important experiments but are currently rather preliminary. For example, n numbers are low for some fibre types (CM and CC) or absent (CH, CMC, CMHC). Moreover it is not indicated what the threshold is for responsiveness in these experiments. The samples sizes and thresholds should be noted for these experiments, and additional experiments are needed in some cases to reach the appropriate numbers*.

We have recorded from 200 identified cutaneous afferents in this study. To further strengthen the K14-NpHR analysis we have performed more experiments. This data has been incorporated into the revised text and added to Table 2. Some fiber types represent a small proportion of fibers in the skin and thus it is difficult to acquire large numbers without performing many more experiments.

*7) The authors have presented data using a single light pulse applied for durations of around 4 seconds. While analysis of the stimulus intensity versus firing rate is presented, it would also be important to show how stimulus duration and the frequency of light pulses can initiate activity. For example what is the minimum duration and number of pulses that can evoke neuronal activity when comparing direct neuronal stimulation versus keratinocyte stimulation? If such data are currently available, they would be a valuable addition to the study*.

We agree. Unfortunately these data are not yet available.